# Renal Disorders Associated with Waldenström Macroglobulinaemia, IgM MGUS and IgM-Producing B-Cell Lymphoproliferative Disorders

**Guy Pratt** [1,2,*] **, Hannah V. Giles** [1] **and Jennifer H. Pinney** [1]

1   University Hospitals Birmingham NHS Foundation Trust, Birmingham B15 2WB, UK;
    hannah.giles@uhb.nhs.uk (H.V.G.); jennifer.pinney@uhb.nhs.uk (J.H.P.)
2   Institute of Cancer and Genomic Sciences, University of Birmingham, Birmingham B15 2TT, UK
*   Correspondence: guy.pratt@uhb.nhs.uk

**Abstract:** Renal disorders are uncommonly associated with IgM MGUS and Waldenström macroglobulinaemia (WM). Data are limited to large case series that suggest that related renal involvement occurs in 5% of patients with WM. Although uncommon, there is a much greater variety of renal pathologies associated with WM and IgM MGUS than that seen in patients with multiple myeloma, where cast nephropathy predominates. In WM, uncommonly direct infiltration of the renal system by lymphoma or cast nephropathy with a high light-chain level can occur. AL amyloidosis can present with nephrotic syndrome as a feature with IgM MGUS or WM. Cryoglobulinaemia and light-chain deposition disease are other important potential causes of renal impairment with IgM MGUS and WM. There are other rarer monoclonal gammopathy of renal significance (MGRS) conditions characterised by typically isolated kidney disease that are causally related to a B-cell or plasma-cell clonal disorder usually in a precancerous MGUS state, although in some renal pathologies, the association is less clear. Central to the majority of these diagnoses is the need for an accurate renal histological diagnosis, and management requires close joint working of renal and haematology teams.

**Keywords:** Waldenströms; MGUS; amyloidosis; cryoglobulin

## 1. Introduction

Waldenström macroglobulinaemia (WM) is a B-cell lymphoproliferative disorder characterized by an immunoglobulin IgM monoclonal protein and bone marrow infiltration by lymphoplasmacytic lymphoma (LPL) [1,2]. A precursor condition IgM monoclonal gammopathy of undetermined significance (MGUS) is defined by the presence of a serum monoclonal IgM of less than 30 g/L, absence of lymphoplasmacytic bone marrow infiltration (according to the 2016 WHO classification, bone marrow lymphoplasmacytic infiltration is <10% in IgM-MGUS) and no other features of WM [1]. Symptomatic WM and IgM MGUS form a spectrum with an intermediate asymptomatic WM stage that can either behave more like an MGUS or progress over time towards symptomatic WM.

Although rare, both WM and IgM MGUS can be associated with a wide variety of renal disorders. Several centres have described retrospective case series. The Mayo Group reported 1363 patients with WM or B-cell lymphoproliferative disorders, where 57 (4%) of these patients had both a bone marrow biopsy and a kidney biopsy [3]. In the large retrospective series by the Dana Farber Institute, Boston, there were a total of 52 (3%) patients in a cohort of 1391 WM patients that had a kidney biopsy performed [4]. Based on these studies, less than 5% of WM patients are likely to have related renal complications. Rarely, IgM monoclonal proteins leading to MGRS renal disorders can also be seen in the context of other low-grade B-cell lymphoproliferative disorders such as chronic lymphocytic leukaemia. The true spectrum, incidence, biology and impact of renal disorders associated with WM and IgM MGUS is poorly understood due to the rare prevalence of renal disorders

in an already uncommon condition. Data are limited to published case reports and large centre retrospective case series [3–7]. The spectrum of renal disorders is much broader than that seen in patients with multiple myeloma, which is predominantly due to cast nephropathy [3–7]. There is greater possibility of an unrelated condition or a condition where the relationship to the plasma-cell disorder is less clear. These diagnoses typically rely on renal histology. The decision to recommend a renal biopsy may be especially difficult when the benefit may not obviously outweigh the risks involved. This is particularly the case when a biopsy result does not alter treatment and when the diagnosis may seem less probable when co-existent common pathologies such as diabetes and hypertension could be the cause of renal impairment.

There are distinct categories of renal disorders associated with WM and IgM MGUS. These cases may be categorised histologically into glomerular disorders (amyloid vs. non-amyloid, such as cryoglobulin-related), tubulointerstitial (typically tumour infiltration or light-chain cast nephropathy), unrelated disorders or those with a poorly understood relationship to WM. Monoclonal gammopathy of renal significance (MGRS) describes kidney disease causally related to a B-cell or plasma-cell clonal disorder in a precancerous MGUS state, and the commonest of these is AL amyloidosis. People with WM can develop the same clonally derived renal lesions as those with an IgM-related MGRS. Renal disorders can occur in WM due to direct effects of the tumour itself, such as tumour infiltration in the kidneys or light-chain cast nephropathy in rare patients with high light-chain levels. Renal pathology that does not appear connected to the plasma-cell disorder or where the relationship is less clear can also be found.

The aim of therapy is to target the B-cell or plasma-cell clone responsible for the renal pathology and improve function before irreversible renal damage occurs. Therefore, the treatment regimens used are identical to those for WM. It is assumed that a greater depth of haematological response ultimately leads to better outcomes, but there is a lack of evidence to help guide treatment. Importantly, complete remissions are difficult to achieve in WM, and this may impact renal responses. Depending on the context, the aim of treatment may mean aiming for independence from dialysis, avoidance of dialysis, improved eGFR and improvement of proteinuria. In patients with relatively preserved renal function, changes in proteinuria/eGFR are critical to monitor and, if deteriorating, may lead to difficult decisions around the timing of treatment.

This review describes the investigation and management of these disorders.

## 2. Investigating Renal Impairment in a Patient with WM or IgM MGUS

The major challenge is defining which patients require a renal biopsy. The individual clinical context for the patient is critical to making a decision regarding a renal biopsy and requires the evaluation of a number of factors, especially the underlying haematological disorder (Figure 1), their past medical history, past renal function, the degree of renal impairment, appropriateness if frail or comorbid and the likely impact on treatment decisions. All patients who have a potential renal-related lesion and IgM monoclonal protein who would benefit from treatment of the underlying clone should have a full haematological workup including bone marrow aspiration and biopsy, and CT scan of neck, thorax, abdomen and pelvis. In addition to routine full blood count and biochemistry tests, the panel of blood tests should include LDH; β2 microglobulin; hepatitis B and hepatitis C; HIV status; serum protein electrophoresis and immunofixation; quantification of IgM, IgG and IgA; serum free light chains; tests that may elucidate the cause for anaemia if haemoglobin is low (haematinics and a haemolysis screen—haptoglobin, lactate dehydrogenase, bilirubin, reticulocyte count, blood film and direct coombs test at 37 °C if cold agglutinin suspected); and importantly, cryoglobulin (mandatory to keep sample at 37 °C). In addition, it is reasonable to rule out acquired Von Willebrand disease in patients with a bleeding history and would be mandatory before a renal biopsy in patients with a significant bleeding history. Testing for von Willebrand disease should also be performed in patients with significantly elevated

IgM (even without a history of bleeding), since acquired von Willebrand syndrome can occur with a high IgM monoclonal protein.

## Investigating suspected IgM related renal disease

**Figure 1.** Recommended investigation for patients found to have an IgM MGUS.

Bone marrow flow cytometry is the standard test used to establish the classical WM B-cell immunophenotype alongside the assessment of MYD88 L265P. Bone marrow trephine biopsies with appropriate immunohistochemistry are essential to characterising the B-cell disorder, particularly as bone marrow infiltration may be patchy.

In patients who do not fulfil the criteria for the treatment of the underlying haematological condition, careful consideration should be given to the requirement for a renal biopsy. Urinalysis is important to detect proteinuria and evaluate if there is microscopic haematuria, which may indicate intrinsic renal pathology. If there is evidence of proteinuria on dipstick, this should be further evaluated by sending urine to measure the albumin-to-creatinine ratio or protein-to-creatinine ratio. Heavy proteinuria of more than 500 mg/24 h or equivalent (ACR > 70 mg/mmol/PCR > 100 mg/mmol) would usually warrant further evaluation with a renal biopsy. In patients with evidence of increasing proteinuria or declining renal function on serial tests, consideration should be given to a renal biopsy. In patients where there is low-level proteinuria and/or microscopic haematuria, and evidence of other potential end-organ damage such as rash or a detectable cryoglobulin, a lower threshold for a renal biopsy may help to provide the required evidence of end-organ damage to enable the treatment of the underlying haematological disease. Skin biopsies have been used to diagnose cryo-/crystalcryoglobuliaemia, where extrarenal involvement is common (purpuric rash, skin ulcers, peripheral neuropathy, and arthralgias).

Renal biopsies must be examined by a specialist renal pathologist, following a standard protocol, with access to all appropriate immunohistochemistry, immunofluorescence and electron microscopy as stated in the tissue pathway for native medical renal biopsies [8]. In the case of AL amyloidosis, a diagnosis can be reached on tissues other than the kidney, such as subcutaneous fat, heart, gut, and bone marrow. Apple-green birefringence on Congo red

staining confirms the diagnosis of amyloidosis. Immunohistochemistry for light chains, Amyloid A, transthyretin, etc., indicates the type of amyloid. Mass spectrometry (laser-capture microdissection with liquid chromatography-coupled tandem mass spectrometry) may be needed to confirm the amyloid subtype in cases where the immunohistochemistry stains are not clear. In the UK, it is recommended to involve the National Amyloid Centre in the diagnosis, as illustrated by high false positive and negative rates of detection of amyloids in non-specialist laboratories.

## 3. Patients with Symptomatic WM with a Significant Tumour Burden and Renal Impairment

A patient with confirmed and symptomatic WM with renal involvement is most likely to have tubulointerstitial infiltration or cast nephropathy. In this scenario a reason for renal impairment may be obvious with high serum free light chain >500 mg/L (cast nephropathy likely) or with tumour involvement of the renal system on CT imaging. However, a cause for renal impairment may not be obvious at all, in which case there is a difficult decision around the need for a renal biopsy, especially as it may not alter treatment.

In the large retrospective series by the Mayo Group [3], only 8/57 (14.0%) patients who had a renal biopsy had evidence of direct tumour effects, and all of these were tubulointerstitial lesions (4/57 (7.0%) lymphoma infiltration alone, 2/57 (3.5%) light-chain cast nephropathy, 1/57 (1.8%) light-chain cast nephropathy with lymphoma infiltration and 1/57 (1.8%) lymphoma infiltration with ANCA-associated glomerulonephritis). In the large retrospective series by the Dana Farber Institute [4], Boston, only 12/52 (23.1%) renal biopsy reports described direct involvement by LPL (8/52 (15.4%) cases were due to LPL infiltration, and 4/52 (7.7%) cases were due to light-chain cast nephropathy). In the French series of 35 patients from eight French centres, they found cast nephropathy in 5/35 (14.3%) patients and isolated tumour infiltration in 1/35 (2.9%) patients [5].

It is essential in this scenario to check serum free-light-chain levels and cryoglobulin and to look for clinical evidence of AL amyloid (see below). A CT scan may show lymphomatous disease in the kidney and/or renal tract.

Treatment may be urgently required if the aim is to reverse renal failure. Treatment is directed at the underlying WM clone and follows national guidelines. Bendamustine–rituximab, dexamethasone–rituximab–cyclophosphamide or a Bruton's tyrosine kinase inhibitor (ibrutinib, zanubrutinib, or acalabrutinib) are all commonly used regimens [9]. Bortezomib regimens (bortezomib, cyclophosphamide rituximab (BCR) and bortezomib, and dexamethasone and rituximab (BDR)) are also effective. Other proteasome inhibitors, such as carfilzomib and ixazomib, are also effective and lack the neurotoxicity seen with bortezomib. Cyclophosphamide requires dose adjustment in advanced renal failure, and there are limited data on Bendamustine in advanced renal failure, so dose reduction should be considered.

## 4. Patients with Suspected Renal Amyloidosis

Typically, these patients present with proteinuria, often to the extent of nephrotic syndrome (more than 3 g/24 h of proteinuria with low serum albumin and peripheral oedema). In the Mayo series [3], 19/1363 (1.4%) patients with WM or B-cell lymphoproliferative disorders had confirmed AL amyloidosis on renal biopsy. The Dana Farber series-reported AL amyloid on renal biopsy was only identified in 11 (0.8%) patients out of a cohort of 1391 WM patients [4]. In the French series of 35 patients, they found amyloids in 11/35 (31.4%) patients [5].

Most commonly, these patients have a lambda light chain (however, 15–20% have a kappa light chain) and an underlying MGUS, but patients can also have more advanced WM. Renal impairment is not usually a major feature, but occasional patients have more advanced renal failure requiring renal support. Multiple-organ involvement is common in AL amyloidosis, so it is critical to assess other features of amyloidosis, namely, heart failure (unexplained raised proBNP/troponin), easy bruising (in particular periorbital bruis-

ing), peripheral neuropathy, macroglossia, gut symptoms/bleeding and liver involvement (raised Alkaline Phosphatase and/or hepatomegaly).

Patients with AL amyloidosis due to an IgM-producing MGUS/WM show differences compared with the commoner AL amyloidosis in the context of a myeloma-type MGUS/MM clone [10]. Patients with an IgM-related AL amyloidosis are reportedly less likely to have cardiac involvement, which is a poor prognostic feature [11]. Despite this, overall, they probably do less well, as complete remission is uncommon, making the regression of amyloidosis less likely [10]. Patients with IgM amyloidosis are more likely to have soft tissue and peripheral nerve involvement and less commonly have the t(11;14) translocation (27% vs. 50%, $p = 0.008$), which is associated with a poorer response to bortezomib-based regimens [10].

Cardiac imaging with echocardiography and cardiac MRI (CMR) are critical in cardiac amyloid assessment in conjunction with cardiac enzymes. CMR requires specific imaging sequences to look at the extracellular volume (which is expanded in amyloidosis) and late gadolinium enhancement.

A tissue diagnosis is essential to confirming the diagnosis of AL amyloidosis. Biopsy of the affected organ is most likely to aid diagnosis, and as renal involvement is a feature in around 50% of cases, this is often the tissue of choice. It is, however, not necessary if amyloids have already been identified in an alternative site such as bone marrow, fat biopsy or gut biopsy. It is important to confirm the type of amyloid as being AL in nature, and this may require mass spectrometry testing (laser-capture microdissection with liquid chromatography-coupled tandem mass spectrometry). Other types of amyloids, such as AA and fibrinogen-related amyloids, can target the kidney, and in this context, the MGUS may be unrelated and incidental. This, therefore, requires a multidisciplinary approach involving cardiologists, pathologists and an amyloid-specialized centre.

Treatment is directed at the underlying WM clone and follows national guidelines as previously discussed. There are no studies to guide the choice of regimen in the context of AL amyloidosis, but the bendamustine–rituximab regimen is the most likely to produce a rapid fall in the pathogenic light chain. Response assessment after each cycle is critical, as suboptimal response after three cycles should prompt consideration around stopping or changing treatment. Complete remission is less common in IgM amyloidosis with lower complete hematologic response rates (6 months: 39% vs. 59%, $p = 0.008$), and as a result, overall survival is shorter in IgM AL amyloidosis when stratified by Mayo 2012 stage (stages 1/2 (59 vs. 125.9 months, $p = 0.003$) and stages 3/4 (6.5 vs. 12.9 months, $p = 0.075$)) compared with non-IgM amyloidosis [10]. In suitable patients, autologous stem-cell transplant (ASCT) may be appropriate and could potentially lead to long treatment-free remission with a deep haematological response that may be difficult to obtain with other treatments. Unfortunately, many patients with AL amyloidosis are excluded from ASCT due to poor fitness, significant cardiac involvement, low GFR, severe autonomic neuropathy, multiple-organ involvement and other high-risk factors such as gut bleeding.

Supportive management in nephrotic syndrome requires close nephrology input and includes careful fluid balance assessment, fluid restriction, and diuretics, which are often required in combination and at high doses to achieve diuresis. Patients with AL amyloidosis and nephrotic syndrome often have low blood pressure and may not tolerate the use of renin–angiotensin system blockage in the early stages, as this may cause profound hypotension. Consideration should be given to thrombotic and bleeding risk in patients with nephrotic syndrome and prophylactic anticoagulation may be required.

## 5. Patients with a Non-Amyloid form of MGRS and Serum Monoclonal IgM

There is a long list of possible IgM-related MGRSs (Table 1). All of these are rare, but the commonest is cryoglobulinaemia [3–7].

**Table 1.** Renal diseases associated with WM/IgM MGUS/B-cell lymphoproliferative disorder.

| Disease/Renal Lesion | Clinical Symptoms | Renal Histology |
|---|---|---|
| Amyloidosis (AL, AH and AHL) | **Renal:** Proteinuria, low serum albumin, oedema and reduced eGFR<br>**Cardiac:** Restrictive cardiomyopathy, shortness of breath, raised cardiac bio-markers<br>Peripheral and/or autonomic neuropathy<br>Easy bruising (peri orbital) and soft tissue infiltration (macroglossia, muscle stiffness and jaw claudication) | Organised fibrillar deposits<br>LM and Congo red-positive deposits<br>IF and IHC-positive deposits for kappa or lambda<br>EM: randomly oriented fibrils (7–14 nm) |
| Cryoglobulinaemia/ Cryoglobulinaemic glomerulonephropathy | **Renal:** Isolated proteinuria or haematuria, and occasionally, nephrotic syndrome<br>**Cutaneous disease:** Digital ischemia, Livedo reticularis and skin necrosis, erythematous macules and purpuric papules in lower limbs, and Reynaud's phenomena<br>**Other manifestations:**<br>Peripheral neuropathy<br>Hyperviscosity symptoms (blurred vision, headache, vertigo, nystagmus, confusion, stroke and coma)<br>Arthralgia, fatigue and myalgia<br>Pulmonary involvement: small airways disease, dyspnoea, cough and pleurisy<br>Lymphadenopathy and splenomegaly<br>Vasculitis of most internal organs possible | Organized microtubular or crystal deposits<br>LM: membranoproliferative or endocapillary proliferative GN with monocyte infiltration and often immune thrombi<br>IF: monotypic Ig granular deposits in glomeruli and vessels<br>EM: microtubular extracellular electron-dense deposits and occasional intracellular crystals |
| Monoclonal immunoglobulin deposition disease (MIDD) LCDD, LHCDD and HCDD | **Predominantly renal-limited** (nephrotic syndrome and common hypertension)<br>Cardiac involvement, cardiomyopathy and heart failure symptoms (rare)<br>Liver involvement<br>Peripheral nerve involvement possible | Non-organised deposits<br>LM: nodular glomerulosclerosis (67% of cases) and thickened TBM<br>IF: linear deposits along GBM, TBM and vessels<br>EM: punctate electron-dense deposits along GBM, TBM and vessels |
| Immunotactoid glomerulonephritis | **Renal isolated disease**<br>Proteinuria seen in 100%<br>Haematuria is common<br>Reduced eGFR in 50% of cases<br>Low complements may be seen | Organised non-fibrillar deposits<br>LM: atypical membranous, membranoproliferative, mesangial or endocapillary proliferative GN<br>IF: monotypic IgG granular deposits in mesangium and GBM<br>EM: glomerular microtubular deposits (14–60 nm) in parallel arrangement |
| Proliferative glomerulonephritis with monoclonal Ig deposits (PGNMID) | **Renal isolated disease**<br>Nephrotic syndrome (50%)<br>Haematuria (70%)<br>Reduced eGFR (70%) | Non-organised deposits<br>LM: membranoproliferative, endocapillary proliferative or membranous GN<br>IF: monotypic Ig (mostly IgG3κ) granular deposits in mesangium and GBM<br>EM: electron-dense deposits in glomeruli |
| Light-chain Proximal Tubulopathy | **Renal isolated disease**<br>Fanconi syndrome, hypophosphataemia, renal glucosuria and/or aminoaciduria<br>Muscle weakness, increased urine output and thirst | Organized crystals or inclusions<br>LM: proximal tubular swelling<br>IF: proximal tubular staining with κ (in crystalline variant) or λ (mostly in non-crystalline variant)<br>EM: proximal tubular LC crystals or lysosomal inclusions |

**Table 1.** *Cont.*

| Disease/Renal Lesion | Clinical Symptoms | Renal Histology |
|---|---|---|
| Fibrillary GN with monoclonal gammopathy | **Renal isolated disease** Proteinuria is common, and nephrotic syndrome is reported in 70% Haematuria is common Reduced eGFR in 50% of cases Low complements may be seen | Organised fibrillar deposits LM: MPGN, diffuse proliferative, mesangial proliferative or diffuse sclerosis IF: Smudged deposits; stain for IgG, C3, κ, or λ, and DNAJB9 EM: Randomly arranged fibrils (12–24 nm) |
| C3 glomerulopathy with monoclonal gammopathy | **Renal isolated disease** Hypertension Proteinuria Declining renal function Low complements in some patients | Non-organised deposits LM: membranoproliferative, mesangioproliferative or endocapillary proliferative GN IF: C3 granular deposits in mesangium and GBM (with paucity of Ig deposits) EM: ill-defined electron-dense deposits in C3GN and intramembranous and mesangial highly electron-dense deposits in dense-deposit disease |
| Thrombotic microangiopathy (TMA) | Unexplained anaemia and thrombocytopenia Hypertension, possibly severe | Light and electron microscopy: GBM duplication, mesangiolysis, subendothelial "fluff," and thrombosis IF: no Ig deposits |
| Membranous nephropathy | **Renal isolated disease** Nephrotic syndrome is common Proteinuria is the predominant feature | LM: diffuse thickening of the GBM. "Spikes" of GBM extending among immune deposits may be seen on silver stain EM: subepithelial electron-dense deposits; effacement of foot processes and expansion of the GBM with deposition of extracellular matrix. IF: diffuse granular pattern of immunoglobulin (Ig) G and C3 staining along the GBM |
| Minimal change nephropathy | **Renal isolated disease** Nephrotic syndrome with rapid onset | LM: glomeruli appear normal EM: diffuse effacement of epithelial foot processes IF: no complement or immunoglobulin deposits |

As with any patient, the likely benefits of treatment have to be weighed against the risk of toxicity, especially in frail, comorbid patients who are less likely to tolerate treatment, and the intensity of treatment should be tailored accordingly.

Treatment is indicated to prevent or slow progression to end-stage kidney disease (ESKD), and in diseases where end-organ damage is confined to the kidneys, there is no benefit from pursuing treatment once a patient has already reached dialysis. Consideration needs to be given to whether such patients with ESKD would be suitable for renal transplant in patients with an MGRS with isolated renal disease. In that scenario, the aim of treatment would be to reduce the pathogenic monoclonal protein prior to renal transplant to avoid disease recurrence in the transplanted kidney.

The measurements of outcomes in MGRS often adopt International Myeloma Working Group or International Consensus AL-Amyloid response criteria, but this may not be optimal or even appropriate for all MGRS. The evaluation of serum free-light-chain levels and ratio results can be particularly challenging in patients with reduced GFR, especially in patients who present with minimally elevated involved free-light-chain levels with subtle free-light-chain abnormalities. Several different modified renal reference ranges have been proposed, but none have been universally adopted [12–14].

Expertise in nephrology and haematology is essential to patient management, and follow-up requires regular evaluation of the haematological response, renal function and proteinuria.

## 6. Cryoglobulinaemic Glomerulopathy

Cryoglobulinaemia is a clinical disorder where immunoglobulins precipitate in vitro at <37 °C and redissolve on heating. The process of precipitation is not well understood, and there is variation in the pathogenicity of cryoglobulin subtypes, varying from asymptomatic to multisystem cryoglobulinaemic vasculitis. Asymptomatic or mild cases may be managed with patient education around avoiding cold exposure. Pathogenicity is related to the individual properties of the protein rather than the level of IgM monoclonal proteins. IgM monoclonal proteins can be associated with type I and type II cryoglobulinemia. There is a serious issue of missing cryoglobulins in the laboratory, as detection critically requires samples to be taken into prewarmed tubes, which must be kept at 37 °C until the serum is separated. The true incidence of cryoglobulinaemia associated with IgM monoclonal proteins is underestimated, as it is not routinely screened for. In a series of 1600 patients with cryoglobulinaemia, 9% of patients had type I cryoglobulinaemia, and 47% had type II cryoglobulinaemia [15].

Cryoglobulinaemic glomerulonephropathy (CGN) is the commonest non-amyloid glomerulonephropathy in WM. In the Mayo series [3], CGN accounted for 12/57 (21.1%) patients with WM or B-cell lymphoproliferative disorders who had both a bone marrow biopsy and a kidney biopsy for review from a total of 1363 patients. Similarly, in the Dana Farber series, 10/52 (19.2%) of WM patients that had a kidney biopsy performed had CGN [4]. The largest and most recent series of 81 patients with non-hepatitis-associated CGN by the Mayo Group found a haematological diagnosis in 89% of patients, including MGRS in 65% and symptomatic lymphoproliferative disorders in 35%. Skin involvement was also present in 64% of patients. In patients with an IgM monoclonal protein, the type of cryoglobulin was type II in 56%, type I in 29% and type III cryoglobulin in 8% [16].

Renal involvement is probably uncommon in type I cryoglobulinaemia, having occurred in only 4% of patients in a small case series [17], and the main features tend to be skin involvement, peripheral neuropathy (or mononeuritis) and arthralgia. In type 1 CGN, the characteristic lesion on light microscopy is a mesangial proliferative glomerulonephritis (MPGN) with endocapillary proliferation. For type I cryoglobulinaemia, treatment should be targeted at the underlying B-cell clone as described in previous sections.

In type II cryoglobulinaemia, a complex is formed between polyclonal heavy chains and monoclonal immunoglobulin (70% of these cases are hepatitis C virus-related). A total of 10–30% of cases are related to a lymphoproliferative disorder, and treatment in these cases is that of the underlying lymphoproliferative disorder. There are limited data specifically relating to cases with type II cryoglobulinaemia with an IgM monoclonal protein, but a case series of non-infectious mixed cryoglobulinaemia suggested that renal involvement was commoner than in type I and that skin, peripheral neuropathy and arthralgia were also common [17]. In the large Mayo series of non-hepatitis-associated CGN, 77% had partial or complete remission, while 10% progressed to kidney failure, and 14% died after a median follow-up of 33 months [16]. Most treatments are clone-directed treatments as used for WM, but steroids and plasma exchange (maintaining fluids at 37 °C) have also been widely used [18]. There is some concern with using rituximab, as a rise in the IgM monoclonal protein, the so-called IgM flare, can worsen symptoms [18]. A minority of cases of non-hepatitis-associated CGN are not associated with a measurable monoclonal protein, presenting with mainly renal isolated disease, and remain poorly understood [16]. Evaluation of serum samples using more sensitive assays for the detection of monoclonal proteins, such as mass spectrometry and modified immunofixation electrophoresis [19–21], may help us understand whether there is a very low-level monoclonal protein that is below the limit of detection of the serological techniques currently used for the detection and monitoring of monoclonal proteins.

## 7. Light-Chain Deposition Disease (LCDD)

Light-chain deposition disease (LCDD) is a rare disease typically presenting with progressive renal disease due to monoclonal immunoglobulin molecules in glomerular and tubular basement membranes with some degree of proteinuria and, less often, heart and liver involvement. Clinical evidence from randomised controlled trials is lacking, and case series report variable extra-renal disease. LCDD is mostly seen in the context of MGUS or symptomatic multiple myeloma, but it has rarely been described in the context of an WM-type IgM MGUS, with one patient being reported in the Mayo series [3] and four cases being reported in the Boston series [4]. The UK National Amyloid Centre reported the presenting features and outcomes of 77 patients with LCDD presenting over a decade, and only 5/77 (6.5%) patients had a serum monoclonal IgM, with 2/5 (40%) of these having WM. In the whole cohort of LCDD patients, 11/77 patients (14.3%) had cardiac involvement, and 10/77 (13%) patients had liver involvement; further, 54/77 patients (70.1%) had eGFR < 30 mL/min, and 10/77 patients (13%) had proteinuria < 0.5 gm/24 h; finally, 8/77 patients (10.4%) had ESRD on dialysis at diagnosis. The median difference between the involved and uninvolved free light chains and bone marrow plasma cells were 450.8 mg/L (range of 2.4–25,690 mg/L) and 8% (range of 1–70%), respectively [22]. Specific data relating to the five patients with LCDD and a serum monoclonal IgM were not presented.

The main indication for treatment in LCDD is preservation of renal function. In patients with advanced kidney disease or those on renal replacement therapy without salvageable renal function, haematological treatment should only be considered in the prospect of kidney transplantation or if there is evidence of organ-threatening extra-renal LCDD. Elevated NT-proBNP, diastolic dysfunction, heart failure and atrial fibrillation may indicate cardiac involvement.

Treatment of associated IgM-producing clone should be based on treatments for WM, bearing in mind that treatment goals are around the reversal of organ dysfunction. Deep haematological response, such as a very good partial response or better, are associated with improved outcomes [22]. Given the challenges to obtaining deep responses in WM and IgM MGUS, there may be a role for ASCT in suitable patients. In the UK NAC series, 23/69 (33.3%) patients underwent autologous stem-cell transplant (ASCT), and 7/77 (9.1%) patients received renal transplant [22]. Prognostic factors include eGFR at baseline, haematological response, extra-renal LCCD (particularly cardiac disease) and the nature of the underlying plasma-cell disorder (MGUS compared with WM).

## 8. Rarer MGRS

All case series [3–7] report small numbers of cases of rarer forms of MGRS. The Mayo Group described two cases of immunotactoid glomerulonephritis, one case of proliferative glomerulonephritis with monoclonal immunoglobulin deposits and one case with more than one pathological feature with both MPGN with minimal change disease and lymphoma infiltration. In the French series of 35 patients, 9/35 (25.7%) were found to have an MPGN [5], but this was not reported in the Boston series [4]. A Chinese series reported one case with C3 glomerulonephritis [7].

## 9. Other Renal Pathologies with Less Clear Relationships to WM and IgM MGUS

A number of other renal pathologies have a less clear relationship with WM and IgM MGUS, such as thrombotic microangiopathy (TMA) (three patients in the Boston series and one patient in the Mayo series—of note, these patients had isolated renal disease), membranous glomerulopathy (one patient in the Boston series) and minimal change disease (MCD) (two patients in the Boston series and two patients in the Mayo series) [3,4]. There is evidence to support a causal relationship in these renal conditions, and it has been described in a range of lymphoproliferative conditions, including WM, multiple myeloma, chronic lymphocytic leukaemia (CLL), and Hodgkin and non-Hodgkin lymphomas [22–26].

TMA may be limited to the kidney or part of a wider microangiopathic haemolytic anaemia (MAHA). Abnormalities in the vessel wall of arterioles and capillaries lead to microvascular thrombosis, and this diagnosis, as with other forms of MGRS, is made on renal biopsy. There are many causes of a TMA, including severe hypertension, systemic infection, autoimmune conditions, and a variety of malignancies in which WM and IgM MGUS are in the differential and should be part of the initial investigations depending on the context of the patients' presentation.

Membranous nephropathy (MN) is one of the most common causes of nephrotic syndrome, accounting for up to one-third of biopsied cases of nephrotic syndrome. MN is most often primary, and it has been associated with hepatitis B, autoimmune diseases, thyroiditis, drugs such as NSAIDs and malignancies. Up to 5–20% of cases in individuals over the age of 65 years have been reported to be associated with a malignancy, the diagnosis of which may not have yet been made, and MN may be part of a paraneoplastic phenomenon. The malignancy may present at the time of presentation of nephrotic syndrome or at a later date. Haematological malignancies such as CLL WM and IgM MGUS have all been reported as being associated with MN [23,27]. There are high rates of spontaneous remission of nephrotic syndrome in MN; therefore, proving benefit from treatment of an associated malignancy is less clear.

Minimal change disease (MCD) accounts for an estimated 10–15% of nephrotic syndrome cases in adults, and haematological malignancies such as lymphoma and leukaemia have been described with MCD and rarely with WM, as described in the Mayo series [3]. By definition, the renal biopsy looks normal on light microscopy, and there is podocyte effacement on electron microscopy. Most cases of MCD respond to treatment with steroids, and in steroid-responsive cases, there is a favourable effect by rituximab [28]. This has led to the suggestion of a role of B cells in addition to T cells in the pathogenesis of MCD. Patients who have an associated WM should, therefore, have rituximab as the backbone of their treatment regimen. Where there is a lack of response to treatment or changes in renal function, a second biopsy may be required, as there have been reported cases of MCD that were later found to be AL amyloidosis [29].

A subtype of MCD is IgM nephropathy. IgM deposits may be found in patients with MCD, focal segmental glomerulosclerosis and MPGN. Electron-dense deposits are seen on electron microscopy, and IgM deposition is also seen on immunofluorescence microscopy. The presence of IgM in a patient with MCD signifies a poorer renal prognosis. When polyclonal IgM deposits are seen, there is not usually a link to a monoclonal IgM haematological disease [30].

## 10. Conclusions and Future Directions

Although rare, the awareness and understanding of IgM MGUS and WM-related renal disorders are important to lead to an optimal diagnosis and management. There are major challenges given the lack of quality data around these conditions and the need for renal histology, which can be challenging, and often, there is lack of clarity around the benefits of treatments. Given this, there is a need for a national WM expert MDT and the use of registry data such as the UK Rory Morrison and rare renal disorders registries (RaDaR) to better characterise and manage these disorders.

**Author Contributions:** G.P., H.V.G. and J.H.P. wrote, reviewed and edited the article. All authors have read and agreed to the published version of the manuscript.

**Funding:** This research received no external funding.

**Institutional Review Board Statement:** Not applicable.

**Informed Consent Statement:** Not applicable.

**Data Availability Statement:** Not applicable.

**Conflicts of Interest:** The authors declare no conflict of interest.

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
