# Peer review of "Renal Disorders Associated with Waldenström Macroglobulinaemia, IgM MGUS and IgM-Producing B-Cell Lymphoproliferative Disorders"

_hemato, doi:10.3390/hemato4020015_

Round 1

Reviewer 1 Report

I thank the authors for their effort to describe the uncommon renal disorders associated with IgM monoclonal gammopathies and B cell  lymphoproliferative disorders. In my opinion, the manuscript is well written but I still have some concerns that should be addressed to make it acceptable for publication in Hemato.

Section 1 (Introduction):

- In the first paragraph, please consider to point out that, according to the 2016 WHO classification, bone marrow lymphoplasmacytic infiltration is < 10% in IgM-MGUS.

Section 4 (The patient with suspected renal amyloidosis):

-  Page 5, line 194: The reference to the UK National Amyloid Centre might be replaced by “an amyloid specialized center” (not only for the readers from the UK).

Section 5 (The patient with a non-amyloid form of MGRS and IgM paraprotein):

Table 1, second row: Regarding clinical sypmtoms, under Raynaud’s phenomena, it might be more understandable to include a subtitle named “Other manifestations” (at the same level of “Cutaneous disease”) that would include peripheral neuropathy (INSTEAD OF nephromathy!), hyperviscosity, etc

-  Misspelling  in Table 1, 10th row: “Renal isolated disease” instead of “Real isolated disease".

Section 7 (Light Chain Deposition Disease (LCDD)):

-  Line 297-298: It is stated that “LCDD is mostly seen in the context of myeloma type MGUS or symptomatic multiple myeloma” while it shoud state “LCDD is mostly seen in the context of MGUS or symptomatic multiple myeloma”.

-  Lines 303-308: In my opinion, it might be more interesting to describe the characteristics of the 5 patients with IgM monoclonal paraprotein instead of the whole series (77 patients) with LCDD reported by the UK National Amyloid Center (Ravichandran et al, Am J Hematol 2022).

Author Response

REVIEWER ONE

Section 1 (Introduction):

COMMENTS 

In the first paragraph, please consider to point out that, according to the 2016 WHO classification, bone marrow lymphoplasmacytic infiltration is < 10% in IgM-MGUS.

RESPONSE

Line 29 - We have added at line 29 in the first paragraph: (according to the 2016 WHO classification, bone marrow lymphoplasmacytic infiltration is < 10% in IgM-MGUS)

Section 4 (The patient with suspected renal amyloidosis):

COMMENT  

Page 5, line 194: The reference to the UK National Amyloid Centre might be replaced by “an amyloid specialized center” (not only for the readers from the UK).

RESPONSE

Page 5, lines 201-203: We have changed “UK National Amyloid Centre” to “an amyloid specialized centre”

Section 5 (The patient with a non-amyloid form of MGRS and IgM paraprotein):

COMMENT-  Table 1, second row: Regarding clinical sypmtoms, under Raynaud’s phenomena, it might be more understandable to include a subtitle named “Other manifestations” (at the same level of “Cutaneous disease”) that would include peripheral neuropathy (INSTEAD OF nephromathy!), hyperviscosity, etc

-  Misspelling  in Table 1, 10th row: “Renal isolated disease” instead of “Real isolated disease".

RESPONSE

We have made all the suggested changes to Table 1 -we have added a subtitle “Other manifestations”. We have corrected spelling errors “neuropathy” for “nephromathy” and “renal isolated disease” for “real isolated disease”

Section 7 (Light Chain Deposition Disease (LCDD)):

COMMENT-  Line 297-298: It is stated that “LCDD is mostly seen in the context of myeloma type MGUS or symptomatic multiple myeloma” while it shoud state “LCDD is mostly seen in the context of MGUS or symptomatic multiple myeloma”.

RESPONSE – Lines 306-307: We have changed “LCDD is mostly seen in the context of myeloma type MGUS or symptomatic multiple myeloma” to “LCDD is mostly seen in the context of MGUS or symptomatic multiple myeloma”.

COMMENT -  Lines 303-308: In my opinion, it might be more interesting to describe the characteristics of the 5 patients with IgM monoclonal paraprotein instead of the whole series (77 patients) with LCDD reported by the UK National Amyloid Center (Ravichandran et al, Am J Hematol 2022).

RESPONSE

We agree but unfortunately this information specific to those 5 patients is missing from this publication. It is a correspondence with supplementary information only. We have added a sentence at line 318 as follows- “Specific data relating to the 5 patients with LCDD and an IgM monoclonal paraprotein were not presented”

Reviewer 2 Report

This is a nice review of a rare manifestation of IgM MGUS and WM.  Overall the review is thorough, but there are a few revisions that are suggested as mentioned below.  

"β2 microglobulin, hepatitis B, C, HIV status" should say "β2 microglobulin, hepatitis B and C, HIV status".  This is also true in Figure 1. 

The authors should also mention and cite that von Willebrand testing should be performed in patients with significantly elevated IgM (even without a history of bleeding) since acquired von Willebrand syndrome can occur with high IgM

"tests elucidate the cause for anaemia" should say "tests that may elucidate the cause for anaemia".  Of note, bilirubin and haptoglobin should also be included in this portion since that helps detect hemolysis also.  

In Figure 1 I think the top right box should be fixed because currently it says "chronic kidney disease with declining renal function" but I think that this would also apply to patients without chronic kidney disease who just have declining kidney function

In Figure 1, technically the serum immunofixation is what detects a monoclonal protein, the serum protein electrophoresis quantifies that amount of monoclonal protein.  Many patients (especially those with amyloid) have a normal serum protein electrophoresis since the monoclonal protein burden is so low, but it is detectable on serum immunofixation.  Also FLC does not detect IgM so that should be removed. So the top left box should say "serum immunofixation protein electrophoresis detects an IgM or light chain paraprotein".  

The Blood Tests box of Figure 1 should also include haptoglobin and bilirubin.  

The Blood Tests box should likely be changed to "Blood and Urine Tests" and include urine testing for protein: urine immunofixation electrophoresis, urine protein electrophoresis, and urine testing for albuminuria and hematuria. 

There is a typo in the bottom middle box of Figure 1.  "iantibody" should be "antibody"

Additionally in WM it is generally recommended these days to also test for CXCR4 and TP53 if available.  This should be added to Figure 1 and the section on bone marrow testing. 

fulfil is spelled incorrectly in the paragraph that starts with "in patients who do not fulfil"

In amyloidosis, >500mg albumin in 24 hours is consistent with renal involvement.  This should be considered and potentially added rather than just saying 1g/24 require more workup. 

"can target the kidney and in this contact the MGUS may" should say "can target the kidney and in this context the MGUS may"

"Bortezomib regimens (bortezomib, cyclophosphamide rituximab (BCR) and bortezomib, dexamethasone and rituximab (BDR) are also effective."  This should perhaps say proteasome inhibitor-based regimens are effective, as many people choose to use other proteasome inhibitors such as ixazomib to help decrease risk of neuropathy which is a big issue in patients with IgM pathologies

"None of these are affected by renal impairment (except cyclophosphamide, which requires dose adjustment in advanced renal failure)."  This is not true because bendamustine dosing has to be adjusted in renal failure, although there are small series that report that you can still use it in renal failure.  This information should be added. 

minor grammatical and spelling errors

Author Response

REVIEWER TWO

This is a nice review of a rare manifestation of IgM MGUS and WM.  Overall the review is thorough, but there are a few revisions that are suggested as mentioned below.  

COMMENT

"β2 microglobulin, hepatitis B, C, HIV status" should say "β2 microglobulin, hepatitis B and C, HIV status".  This is also true in Figure 1. 

RESPONSE

Line 87  We have changed "β2 microglobulin, hepatitis B, C, HIV status" to "β2 microglobulin, hepatitis B and C, HIV status". 

Figure One We have changed "β2 microglobulin, hepatitis B, C, HIV status" to "β2 microglobulin, hepatitis B and C, HIV status".  

COMMENT

The authors should also mention and cite that von Willebrand testing should be performed in patients with significantly elevated IgM (even without a history of bleeding) since acquired von Willebrand syndrome can occur with high IgM

RESPONSE

We have added a comment Line 96

“Testing for von Willebrands disease should also be performed in patients with significantly elevated IgM (even without a history of bleeding) since acquired von Willebrand syndrome can occur with a high IgM monoclonal protein.”

COMMENT AND RESPONSE

Line 89 -we have changed "tests elucidate the cause for anaemia" to "tests that may elucidate the cause for anaemia". 

COMMENT

Of note, bilirubin and haptoglobin should also be included in this portion since that helps detect hemolysis also.  

RESPONSE

Line 90 - We have included haptoglobin, lactate dehydrogenase, bilirubin, reticulocyte count, blood film and direct coombs test at 37oC if cold agglutinin suspected

COMMENT

In Figure 1 I think the top right box should be fixed because currently it says "chronic kidney disease with declining renal function" but I think that this would also apply to patients without chronic kidney disease who just have declining kidney function

RESPONSE

Agree -we have changed the top right box to say “patients with declining kidney function or with proteinuria (urine ACR> 30)”

COMMENT

In Figure 1, technically the serum immunofixation is what detects a monoclonal protein, the serum protein electrophoresis quantifies that amount of monoclonal protein.  Many patients (especially those with amyloid) have a normal serum protein electrophoresis since the monoclonal protein burden is so low, but it is detectable on serum immunofixation.  Also FLC does not detect IgM so that should be removed. So the top left box should say "serum immunofixation protein electrophoresis detects an IgM or light chain paraprotein".  

RESPONSE

Agree -in the top left box of Figure 1 we have changed "serum immunofixation protein electrophoresis detects an IgM or light chain paraprotein".  

COMMENT

The Blood Tests box of Figure 1 should also include haptoglobin and bilirubin.  

RESPONSE

In Figure 1 to the Blood tests box - we have added haptoglobin, lactate dehydrogenase, bilirubin, reticulocyte count, blood film and direct coombs test at 37oC if cold agglutinin suspected

COMMENT

The Blood Tests box should likely be changed to "Blood and Urine Tests" and include urine testing for protein: urine immunofixation electrophoresis, urine protein electrophoresis, and urine testing for albuminuria and hematuria. 

RESPONSE

We have changed Blood Tests box to “Blood and urine tests”. We have included urine tests for albuminuria and haematuria.

We do not test urine for urine immunofixation electrophoresis, urine protein electrophoresis in our country and feel that it is not needed so we have left it out. It is expensive and is of questionable value.

COMMENT

There is a typo in the bottom middle box of Figure 1.  "iantibody" should be "antibody"

RESPONSE

In Figure 1 -We have changed  "iantibody" to "antibody"

COMMENT

Additionally in WM it is generally recommended these days to also test for CXCR4 and TP53 if available.  This should be added to Figure 1 and the section on bone marrow testing. 

RESPONSE

Agree – we added these to Figure 1 in the section on bone marrow testing 

COMMENT

fulfil is spelled incorrectly in the paragraph that starts with "in patients who do not fulfil"

RESPONSE

Line 105 We have corrected “fulfil to “fulfill the..”. We have added a comma to that sentence after “condition” - In patients who do not fulfill the criteria for treatment of the underlying haematological condition, careful consideration should be given to the requirement for a renal biopsy.

COMMENT

In amyloidosis, >500mg albumin in 24 hours is consistent with renal involvement.  This should be considered and potentially added rather than just saying 1g/24 require more workup. 

RESPONSE

Line 109 - We have changed 1g/24 hours to 500mg/24 hours

COMMENT

"can target the kidney and in this contact the MGUS may" should say "can target the kidney and in this context the MGUS may"

RESPONSE

Line 200 -we have changed “contact” to “context”

COMMENT

"Bortezomib regimens (bortezomib, cyclophosphamide rituximab (BCR) and bortezomib, dexamethasone and rituximab (BDR) are also effective."  This should perhaps say proteasome inhibitor-based regimens are effective, as many people choose to use other proteasome inhibitors such as ixazomib to help decrease risk of neuropathy which is a big issue in patients with IgM pathologies

REPSONSE

Line 159 We agree – we have added – “Other proteasome inhibitors such as carfilzomib and ixazomib are also effective and lack the neurotoxicity seen with bortezomib.”

COMMENT

"None of these are affected by renal impairment (except cyclophosphamide, which requires dose adjustment in advanced renal failure)."  This is not true because bendamustine dosing has to be adjusted in renal failure, although there are small series that report that you can still use it in renal failure.  This information should be added. 

RESPONSE

Line 161 We agree and therefore have changed -  there is data for bendamustine in renal failure suggesting it is safe with GFR >10ml/min – We have changed the sentence to - Cyclophosphamide requires dose adjustment in advanced renal failure and there is limited data for Bendamustine in advanced renal failure so dose reduction should be considered.